# The Effect of Sleep–Wake Routines on the Negative Emotional States and Aggressive Behaviors in Adults with Autism Spectrum Disorders (ASD) during the COVID-19 Outbreak

**DOI:** 10.3390/ijerph19094957

**Published:** 2022-04-19

**Authors:** Annalisa Levante, Serena Petrocchi, Costanza Colombi, Roberto Keller, Antonio Narzisi, Gabriele Masi, Flavia Lecciso

**Affiliations:** 1Department of History, Society and Human Studies, University of Salento, 73100 Lecce, Italy; flavia.lecciso@unisalento.it; 2Laboratory of Applied Psychology, Department of History, Society and Human Studies, University of Salento, 73100 Lecce, Italy; serena.petrocchi@usi.ch; 3Faculty of Biomedical Sciences, Università della Svizzera Italiana, Via Buffi 13, 6900 Lugano, Switzerland; 4IRCCS Stella Maris Foundation, 56018 Pisa, Italy; costanza.colombi@fsm.unipi.it (C.C.); antonio.narzisi@fsm.unipi.it (A.N.); gabriele.masi@fsm.unipi.it (G.M.); 5Adult Autism Center, Mental Health Department, Local Health Unit ASL Città di Torino, 10138 Turin, Italy; roberto.keller@aslcittaditorino.it

**Keywords:** COVID-19, autism, adult, sleep–wake routines, aggressive behaviors, emotional states, mediation

## Abstract

Disruption in routine may be related to experiencing negative emotional states and to aggressive behaviors in individuals with Autism Spectrum Disorder (ASD). The lockdown because of COVID-19 contributed to the disruption of individuals’ routines, including the sleep–wake cycle. The current study tested a relationship between the adherence to the sleep–wake routine and aggressive behaviors via the mediation role of negative emotional states (i.e., anxiety and anger). Forty-three parents of adults with ASD completed a web-based questionnaire about their life condition during the first lockdown (April–May 2020). Preliminary analyses showed a worsening in the adults’ aggressive behaviors during the lockdown in comparison to before it (Z = −3.130; *p* = 0.002). In the mediation models, the relationship between the adherence to the sleep–wake routines and aggressive behaviors was significant. The models showed the hypothesized mediated relationships among the adherence to the sleep–wake routines, negative emotional states, and aggressive behaviors (Model 1: F _(1, 41)_ = 10.478, *p* < 0.001; Model 2: F_(1, 41)_ = 9.826, *p* = 0.003). The findings confirmed the potential protective role of the adherence to the sleep–wake routines for the emotional and behavioral adjustment of adults with autism. Theoretical and practical contributions of the study were discussed; indeed, our results may inform parent-coaching as well as intervention programs for individuals with ASD given that adequate sleep hygiene may contribute to improvements in internalizing/externalizing behaviors.

## 1. Background

Since the COVID-19 pandemic was declared in January 2020 [1], among all countries, Italy has been one of the most affected. To reduce the spread of the infection, several public health measures were imposed. During the first COVID-19 wave in 2020, mobility was allowed only for primary needs (i.e., food and urgent health requirements) for about three months. Schools and universities as well as recreational and sport centers remained closed for several months. Health care services not directly related to the COVID-19 emergency management were suspended. As a sequela of these extraordinary conditions, a set of psychological detrimental outcomes have been identified in a vast majority of the population. In the general population, high levels of uncertainty [2] contributed to high levels of anxiety, depression, worry, and fear of contagiousness [3,4]. Moreover, some accounts suggest that the lockdowns contributed to the onset of several psychopathological symptoms and emotional distress [5,6,7]. Regarding psychiatric populations, the systematic review by Vindegaard and Benros [4] found that, in general, the clinical conditions and symptoms of individuals worsened during the lockdowns imposed by the COVID-19 sanitary emergency. Nevertheless, this systematic review also highlighted that only a few studies evaluated the impact of the COVID-19 outbreak on vulnerable populations with preexisting disabilities. Therefore, further research is needed to evaluate the impact of COVID-19 on fragile populations including individuals with autism spectrum disorder (ASD). The present study aimed to evaluate the impact of daily routine disruption on the wellbeing of adult individuals with ASD. Specifically, the study contributions were twofold: on one hand, our investigation could extend the knowledge regarding the effect that the lockdown due to COVID-19 had had on these vulnerable families, as well as the pivotal role of routine management; on the other hand, our results could provide knowledge, which could inform interventions aimed at supporting families with sons/daughters with ASD during stressful conditions.

### 1.1. ASD and the Disruption of Routines

According to the DSM-5 [8], one of the main symptoms of ASD’s restricted and repetitive patterns of behaviors include inflexible adherence to routines, which may lead to severe difficulties in adapting to new situations. When the routines of individuals with ASD are disrupted (e.g., therapy-, school-, or sleep–wake related routines), the severity of social and communication deficits and the intensity of repetitive and restricted behaviors may be exacerbated in individuals with ASD. Moreover, routine disruption may lead to the upsurge or worsening of aggressive behaviors in this population [9,10]. Routine disturbance in individuals with ASD has been related to instability and increased fragility in their family environment [11] Therefore, families need support when their members with ASD experience worsening symptoms due to routine disruption.

Among the routines adhered to by individuals with ASD, the sleep–wake one is pivotal for daily functioning in terms of cognitive and social functioning and attention [12,13,14]. Disruption in sleep–wake routines may significantly and negatively affect the severity of the core ASD severity [15,16,17,18,19]. Furthermore, sleep–wake routines’ disruption has been related to emotional and problem behaviors. Research showed that sleep disturbance may be related mood variability [20], as well as problem behaviors including aggression and disruptive behaviors [21,22,23]. To date, there is little empirical research [24,25] examining how sleep–wake routines are related to individuals’ emotional states and aggressive behaviors.

### 1.2. The Interplay between Sleep–Wake Routines, Aggressive Behaviors, and Emotional States

Individuals with ASD are at high risk of presenting aggressive behaviors [26,27,28]. Among the factors that could affect these maladaptive behaviors, disturbed sleep–wake routines may play a major role [16,29,30,31]. The comprehensive theoretical framework by Hollway and Aman [20] suggested a bidirectional causality relationship between externalizing behaviors and sleep problems; that is, an aggressive behavior and the related emotional dysregulation led individuals to have difficulty maintaining the sleep–wake cycle, and the disrupted sleep–wake routine led the individuals to externalize aggressive behaviors toward themselves and others. As noted by Cohen and colleagues’ [10], the relationship between sleep–wake routines and aggressive behaviors has received little empirical attention. Furthermore, except for Matson and colleagues’ [29] study, to date, no one has focused attention on adults with ASD. These authors found that adults with ASD who exhibited moderate to severe sleep disturbance engaged in more aggressive behaviors (e.g., kicking, banging or throwing objects, and yelling at others) in comparison to those with mild sleep problems. Based on this previous evidence, we formulated our first hypothesis:

**Hypothesis** **1** **(H1).**
*We expected to find a negatively increased aggressive behavior in individuals with low adherence to sleep–wake routines. In other words, a lower adherence to the sleep–wake routines would be rated to higher levels of aggressive behaviors (HP1).*


In addition, disturbed sleep–wake routines have been related to individuals’ negative emotional states [32]. In particular, some evidence [33,34] has highlighted that disturbed sleep–wake routines provoked anxiety, anger, and irritability. Moreover, negative emotional states are associated with aggressive behaviors [35]. In terms of this aspect, we formulated our second hypothesis:

**Hypothesis** **2** **(H2).**
*We expected to find a negative impact of the lack of adherence to the sleep–wake routines on the emotional states of adults with ASD. In other words, a lower adherence to the sleep–wake routines would be related to higher levels of anxiety and anger (HP2).*


Both these two negative emotional states were found [36,37,38] to be expressed by individuals with ASD as externalizing, i.e., aggressive behavior, toward themselves or others. Based on this evidence, we formulated our third hypothesis:

**Hypothesis** **3** **(H3).**
*We expected to find a positive relationship between negative emotional states (anxiety and anger) and aggressive behaviors in adults with ASD. In other words, higher negative emotional states lead to higher levels of aggressive behaviors (HP3).*


Although the abovementioned studies are valuable, they considered only children or adolescents with ASD, and they did not analyze the interplay between the considered variables. Therefore, the present research intended to address this gap in the literature testing the mediation effect of the emotional states on the relationship between sleep–wake routines and aggressive behaviors.

## 2. Method

### 2.1. Design and Procedure

Data were collected during the first mandatory lockdown due to COVID-19 in Italy (April–May 2020). An online survey was implemented using Qualtrics XM software and disseminated via the main social media platforms (i.e., WhatsApp, Facebook) via the snowball method. The predefined inclusion criteria were: (1) having a son/daughter with an ASD diagnosis aged over 18 years and (2) the parental ability to read and understand the Italian language. The exclusion criterion was the presence of any preexisting medical condition for the parents who completed the survey. The questions were previously tested on the general population [5,6] and on parents of a child with ASD [39]. The University Ethical Committee approved the research (No. 53162; April 2020), and the study was conducted in accordance with the guidelines of the Declaration of Helsinki. The parents read an information sheet and electronically signed the consent form before their participation. Every participant received a debriefing at the end of the questionnaire.

### 2.2. Participants

Forty-three parents [M(sd) = 54.8(10.2) years] of an adult son/daughter with ASD [M(sd) = 24.8(6) years] completed the online survey. Table 1, Table 2 and Table 3 summarize the sociodemographics of the son/daughter with ASD and their parents, respectively.

### 2.3. Measure

*Adherence to the sleep*–*wake routines*. Two items [5,6,39] asked the parents to say how much their adult children adhered to the sleep- (“*Did your son/daughter adhere to sleep routines during the lockdown?*”) and wake- (“*Did your son/daughter adhere to wake routines during the lockdown?*”) related routines. Response options ranged from 1 (“not at all”) to 5 (“very much”), with high scores indicating higher adherence. An aggregate total score (r = 0.785; *p* < 0.001) was calculated as the average of the two items with higher scores indicating higher adherence.

*Negative emotional states.* Two questions [5,6,39] asked the parents to say how much their adult son/daughter was anxious (“*During the lockdown, how much do you think your son/daughter was anxious?*”) and angry (“*During the lockdown, how much do you think your son/daughter was angry?*”) during the previous week corresponding to the lockdown period. Response options ranged from 1 (“not at all”) to 5 (“very much”), with high scores indicating higher levels of anxiety and anger/irritability.

*Aggressive behaviors*. Two questions [39] asked parents to report how often their adult son/daughter show aggressive behavior toward him/herself (“*Was your son/daughter aggressive towards him/herself during the lockdown?*”) and others (“*Was your son/daughter aggressive towards others during the lockdown?*”). Response options ranged from 0 (“not at all”) to 7 (“very much”), with high scores indicating higher aggressive problems. An aggregate total score (r = 0.339; *p* = 0.026) was calculated as the average of the two items.

*Behavioral problems*. In order to explore behavioral problems in adults with ASD before and during the lockdown, a total of 12 ad hoc questions were formulated. Specifically, six questions asked parents how often—before the lockdown—their adult son/daughter showed a set of behavioral problems, such as: “*How often did your son/daughter show stereotyped behaviors before the lockdown?*”, “*How often did your son/daughter show aggressive behaviors towards him/herself before the lockdown?*”, “*How often did your son/daughter show aggressive behaviors towards others before the lockdown?*”, “*How often did your child show repetitive behaviors before the lockdown?*”, “*How often did your child have difficulties in managing the transition from one activity to another before the lockdown?*”, “*How often did your child show hyper- or hyporeactivity to light/movement activity/touch material before the lockdown?*”). A set of six similar questions were formulated to asked parents how often their son/daughter showed these behavioral problems during the lockdown. Response options ranged from 0 (“not at all”) to 7 (“very much”), with higher scores indicating greater behavioral problems.

### 2.4. Statistical Analyses

The statistical analyses were performed using SPSS version 25 (IBM Corporation, Armonk, NY, USA) [40]. In order to examine the associations between variables, Spearman rho correlations were carried out. Preliminary comparisons between behavioral problems before and during lockdown due to COVID-19 were performed via Wilcoxon Z-tests. Mediation models were performed using Process v3.0 (https://www.processmacro.org/index.html, accessed on 10 June 2021), applying Model 4 and 5000 bootstraps inference for model coefficients. In the mediation models, the predictor variable (x) was the adherence to the sleep–wake routines, the outcome (y) was the aggressive behaviors, and the mediators (M) were the anxious (Model 1) and angry (Model 2) emotional states.

## 3. Results

### 3.1. Preliminary Comparisons before and during the Lockdown

Regarding behavioral problems, the results of the comparison between before and during the lockdown are summarized in Table 4. The findings showed an increase in stereotypes and aggressive behaviors toward the self and others. Additionally, an increase in ritualized patterns of behaviors and in the ability to manage the transition from one activity to another were found. Specifically, parents reported that their adult son/daughter showed more behavioral problems during the mandatory lockdown than before it.

### 3.2. Main Analyses

Spearman’s rho correlations showed that adherence to the sleep–wake routines was significantly and negatively related both to the adults’ aggressive behaviors (rho = −0.705; *p* < 0.001) and their negative emotional states (anxiety: rho = −0.473; *p* = 0.001; anger: rho = −0.438; *p* = 0.003). Furthermore, the adults’ anxiety and anger correlated with aggressive behaviors (anxiety rho = 0.487; *p* = 0.001, anger rho = 0.638; *p* < 0.001).

The two mediation models were both significant (Model 1: F _(1, 41)_ = 10.478, *p* < 0.001; Model 2: F_(1, 41)_ = 9.826, *p* = 0.003). In both models, the two total effects were significant. Table 5 reports the effects.

In both models, the direct paths between the adults’ adherence to the sleep–wake routines and the aggressive behaviors were significant and negative, indicating that a low adherence to the sleep–wake routines led to externalizing more aggressive behaviors toward the self and others. The adults’ adherence to the sleep–wake routines were significantly and negatively related to the negative emotional states of anxiety and anger, which, in turn, were significantly and positively associated with aggressive behaviors. Figure 1 and Figure 2 report the regression coefficients and bootstrap intervals of confidence.

Since the design of the present study is cross-sectional, two alternative reverse models were carried out (Model 3 and 4), in which anxiety and anger were considered as independent variables, the aggressive behaviors as the mediator, and the adherence to the sleep–wake routines as the dependent variable. The two mediation models were both significant (Model 3: F _(1, 41)_ = 16.124, *p* < 0.001; Model 4: F_(1, 41)_ = 37.800, *p* < 0.001) and in both models, the two total effects were significant. Table 6 reports the effects.

In both models, the direct paths between the adults’ negative emotional states, anxiety and anger, and the adherence to the sleep–wake routines were not significant, whereas the two indirect effects were. Figure 3 and Figure 4 report the regression coefficients and bootstrap intervals of confidence. In both models, the adults’ anxiety and anger emotional states were significantly and positively related to the aggressive behaviors, which, in turn, were significantly and negatively associated with adults’ adherence to the sleep–wake routines. Figure 3 and Figure 4 report the regression coefficients and bootstrap intervals of confidence.

## 4. Discussion

A recent systematic review [4] on symptoms worsening in individuals with mental health disabilities related to the COVID-19 sanitary emergency emphasized the paucity of the research on this topic. Thus, our study aimed to extend the knowledge regarding the impact of the lockdown due to COVID-19 on adults with ASD. To be specific, it sought to evaluate the psychological outcomes of individuals with ASD during the first phase of the mandatory lockdown in Italy. To the best of our knowledge, this is the first study addressing the impact of the disruption of the adherence to the sleep–wake routines disruption on aggressive behaviors via the mediation role of the negative emotional states in adults.

Our study investigated problem behaviors in adults with ASD, before and during the first lockdown, as reported by parents. In this vein, the results indicate that, during the lockdown, adults with ASD showed more stereotypes, aggressive behaviors toward self and others, ritualized patterns of behaviors, and more difficulty in transitioning from one activity to the another. To date, no studies on adults with ASD have investigated considering these same variables; nevertheless, similar results were found in children and the adolescent population with ASD [39,41].

The main results of our study included the significant interplay between the adherence to the sleep–wake routines, negative emotional states (anxiety and anger), and aggressive behaviors in adults. Our work tested two mediation models (Model 1 and 2) and two alternative reverse models (Model 3 and 4). The findings regarding Model 1 and 2 showed that poor adherence to the sleep–wake routines was associated with more aggressive behaviors toward self and others, as well as with more anxiety and anger. Furthermore, experiencing negative emotional states was positively associated with externalizing aggressive behaviors. Regarding the reverse models (Model 3 and 4), when the aggressive behavior was used as mediator on the relationship between emotional states and adherence to the sleep–wake routines, the results showed significant indirect effects.

Specifically, regarding the relationships between the adherence to the sleep–wake routines and aggressive behaviors (HP1), our results corroborated the hypothesized associations as theorized in terms of the causal bidirectional link by Hollway and Aman [20]. Regarding the relationship between the adherence to the sleep–wake routines and negative emotional states (HP2), our findings supported the hypothesis and then supported the idea that poor and/or disrupted adherence to the sleep–wake routine leads to behavioral problems, which appear as an additive burden on individuals with ASD behavioral adjustment [33,34]. Regarding our third hypothesis (HP3: effect of anxiety and anger on aggressive behaviors), our findings confirmed the causality bidirectional relationship between these two variables suggested by others [35,36,37]. Albeit preliminarily, the findings regarding the mediation models were promising, and they extended knowledge on this relationship.

Regarding Model 3 and 4, in HP1 as well as in HP3 the reverse relationships were significant. Specifically, the results showed that the aggressive behaviors externalized by the adults with ASD predicted a disruption in the adherence to the sleep–wake routines (HP1 reverse); furthermore, the negative emotional states experienced by the participants predicted the aggressive behaviors (HP3 reverse). Finally, in HP2, the reverse models were not significant. This means that the establishing and/or the maintenance of the adherence to the sleep–wake routines lead the individuals with ASD to experience low levels of anxiety and anger and to externalize less aggressive behaviors. It would be interesting to further explore this reverse relationship in future research.

In sum, the disruption of the adherence to the sleep–wake routines because of the COVID-19 lockdowns could reflect psychological distress experienced by adults with ASD and their parents [42]. This distress could be related to the change in general routines which, in turn, affected the sleep–wake patterns. Finally, these findings are supported by other studies [16,29,30,31,33,34,35,36,37,38], which considered each independent relationship (e.g., adherence to the sleep–wake routine and aggressive behaviors; negative emotional states and aggressive behaviors) but not the interplay between them. Further studies on this topic were required.

## 5. Limitation 

The present study had several limitations. The research had a cross-sectional design; therefore, any conclusion regarding causality cannot be considered. Based on this issue, two models based on the associations found in the literature, and two alternative reverse models were tested. All four models tested demonstrated the hypothesized relationships among variables. Future research with a longitudinal design should disentangle the causal link among the variables. The sample size was small and nonrandom, due to the clinical population recruited during COVID-19 outbreak, and the data were parent-reported. Although the sample size was consistent with other research including ASD individuals [11], and the research supported the validity of the parents’ reports of their adult children routines, future longitudinal studies as well as research devoted to attention to the professionals’ perception are recommended to support the validity of our evidence. Finally, the measures consisted in ad hoc and nonstandardized questions. Although they were tested in previous research [5,6,39], further research is needed in this field. For example, the questions regarding adherence to the sleep-wake routines can be developed in a more detailed way to include specific behaviors that are involved in those routines. This would allow creating a more reliable measure.

## 6. Conclusions

The impact of the lockdowns on the routines of individuals with ASD has been inevitable. Our study demonstrated that the behavioral problems of adults with ASD worsened during the mandatory lockdown due to COVID-19. Moreover, the mediation models demonstrated the hypothesized relationships between the adherence to the adherence to the sleep–wake routines, negative emotional states, and aggressive behaviors. The more the lockdown resulted in a disruption of the routines, the more adults with ASD expressed negative emotions and behavioral problems. The reverse models also demonstrated that the higher the levels of anger and anxiety, the worse the behavioral problems and the disruption of the routines.

This indicates there is a need to further evaluate this topic to define parent-coaching programs and individuals’ intervention by adults. Our results give important insight into how adults with ASD can be supported in the management of adherence to the sleep–wake routines to mitigate the worsening of emotional states and aggressive behaviors during the ongoing COVID-19 outbreak. This could also have a positive impact in the case of future health crises or the familiar unforeseen (e.g., hospitalization or moves).

The results of the present investigation may inform interventions. Parent-coaching programs and specific intervention for adults with ASD on how to manage sleep–wake routines and their distress levels are pivotal. As suggested by Devnani and colleagues [43], bedtime routines (e.g., sleep hygiene; relaxing sleep environment; low exposure to arousal stimuli; low exposure to screen devices; respecting the sleep- and wake-up specific hour), together with pharmacological interventions [44], are essential for individuals with ASD in helping them to establish positive sleep patterns and improve daytime functioning. Those behavioral interventions, based on bedtime routines, should also consider the role played by distress and emotional arousal on the adherence to the sleep–wake routines. This could have a related positive benefit on the management of behavioral problems.

## Figures and Tables

**Figure 1 ijerph-19-04957-f001:**
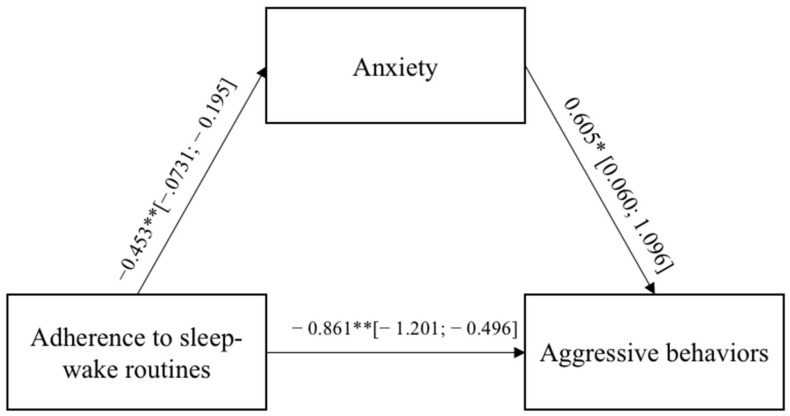
Results of Model 1. The bootstrap intervals of confidence are reported in brackets. * *p* < 0.05; ** *p* < 0.01; *** *p* < 0.001.

**Figure 2 ijerph-19-04957-f002:**
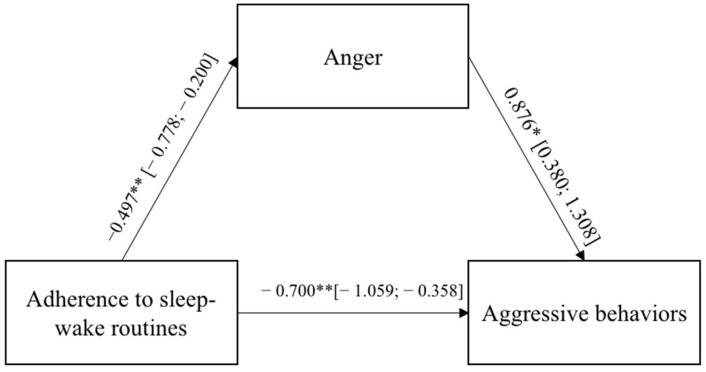
Results of Model 2. The bootstrap intervals of confidence are reported in brackets. * *p* < 0.05; ** *p* < 0.01; *** *p* < 0.001.

**Figure 3 ijerph-19-04957-f003:**
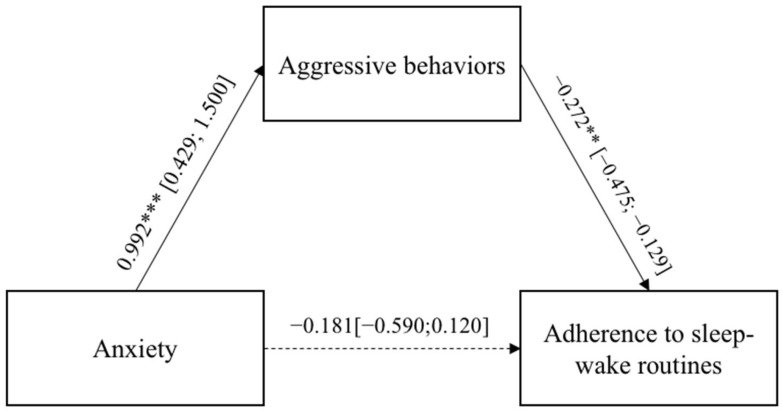
Results of Model 3. The bootstrap intervals of confidence are reported in brackets. * *p* < 0.05; ** *p* < 0.01; *** *p* < 0.001.

**Figure 4 ijerph-19-04957-f004:**
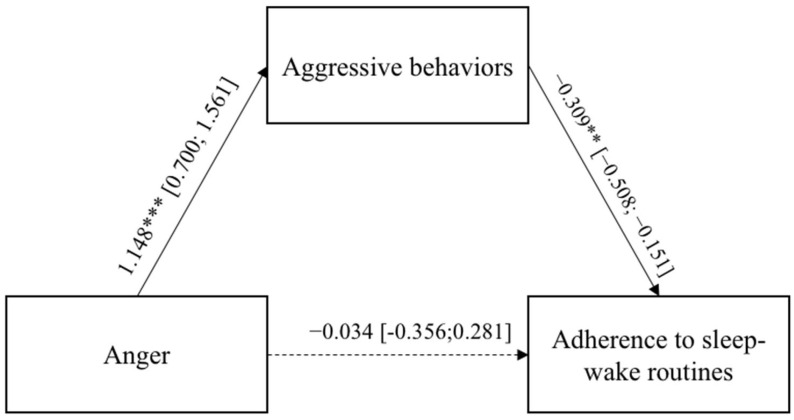
Results of Model 4. The bootstrap intervals of confidence are reported in brackets. * *p* < 0.05; ** *p* < 0.01; *** *p* < 0.001.

**Table 1 ijerph-19-04957-t001:** Characteristics of adults with ASD.

Variable	N (%)
Sex	
Male	33 (76.7%)
Female	10 (23.3%)
Severity of ASD symptomatology	
High-functioning	18 (41.9%)
Low-functioning	25 (58.1%)
Comorbidity ^1^	
Intellectual disability	32 (74.4%)
Language disorder	22 (51.2%)
Oppositional–defiant disorder	12 (27.9)
Obsessive–compulsive disorder	13 (30.2%)
Motor coordination disorder	6 (14%)
Learning disorder	13 (30.4%)
Epilepsy	6 (14%)
Attention deficit and Hyperactivities	16 (37.2%)
Behavioral disorders	17 (39.5%)
Depressive disorder	6 (14%)
Anxious disorder	18 (41.9%)
Psychotic disorder	5 (11.6%)
Hours spent using screen-based devices ^2^ (before the lockdown)	
<2 h/day	25 (58.1%)
2–6 h/day	16 (37.2%)
>6 h/day	2 (4.7%)
Hours spent using screen-based devices (during the lockdown) ^2^	
<2 h/day	12 (27.9%)
2–6 h/day	21 (48.8%)
>6 h/day	10 (23.3%)

^1^ Parent-reported information. ^2^ Adult with ASD-reported information.

**Table 2 ijerph-19-04957-t002:** Parental characteristics.

Variable	N (%)
Sex	
Mother	35 (81.4%)
Father	8 (18.6%)
Marital status	
Without a partner	15 (34.9%)
With a partner	28 (65.1%)
Educational level	
Low (≤8 years of education)	9 (20.9%)
Intermediate (≤13 years of education)	22 (51.1%)
High (≥16 years of education)	12 (28%)
Employment status	
Employed	23 (53.5%)
Not employed	20 (46.5%)
Health profession	
Yes	6 (13.9%)
No	37 (86.1%)
Continuing to work at home	
Yes	20 (46.5%)
No	23 (53.5%)

**Table 3 ijerph-19-04957-t003:** Parent-reported sociodemographic information regarding adults with ASD.

Variable	N (%)
Sport activities practiced by son/daughter with ASD during the lockdown	
Less than usual	5 (11.6%)
As usual	19 (44.2%)
More than usual	2 (4.7%)
Stopped practicing sport	9 (20.9%)
Never practices sport	8 (18.6%)
Medical therapy during the lockdown	
Suspended	1 (2.3%)
As usual	19 (44.2%)
Increased	4 (9.3%)
Psychological therapy before the lockdown	
Yes	12 (27.9%)
No	31 (72.1%)
ASD symptomatology trend during the lockdown	
Enhanced	3 (7%)
As usual	19 (44.2%)
Worsened	21 (48.8%)
ASD behavioral problem trend during the lockdown	
As usual	20 (46.5%)
Worsened	17 (39.5%)
Absent	6 (14%)
Hours per day using screen-based devices (before the lockdown) ^1^	
<2 h/day	25 (58.1%)
2–6 h/day	16 (37.2%)
>6 h/day	2 (4.7%)
Hours per day spent using screen-based devices (during the lockdown) ^1^	
<2 h/day	12 (27.9%)
2–6 h/day	21 (48.8%)
>6 h/day	10 (23.3%)

^1^ Parent-reported information.

**Table 4 ijerph-19-04957-t004:** Comparison between before and during the lockdown in behavioral problems.

Behavioral Problems	Before the Lockdown M(sd)	During the Lockdown M(sd)	Z; *p*
Stereotypes	3.58(1.95)	4.40(2.28);	Z = −3.470; *p* = 0.001
Aggressive behaviors toward the self	1.02(1.42);	1.65(2.21)	Z = −3.130; *p* = 0.002
Aggressive behaviors toward the other	1.16(1.5);	1.67(2.32);	Z = −2.279; *p* = 0.023
Ritualized behaviors	2.70(2.05)	3.67(2.51)	Z = −3.963; *p* < 0.001
Transition from one activity to another	2.21(1.82)	2.81(2.36)	Z = −2.746; *p* = 0.006

**Table 5 ijerph-19-04957-t005:** Results of mediation Models 1 and Model 2.

Direct Effect	Indirect Effect	Total Effect
**Model 1**		
Path: Adherence to sleep–wake routines → Anxiety → Aggressive Behaviors		
β = −0.861 **	β = −0.274 **	β = −1.135 ***
**Model 2**		
Path: Adherence to sleep–wake routines → Anger → Aggressive Behaviors		
β = −0.700 **	β = −0.436 **	β = −1.135 ***

Note: * *p* < 0.05; ** *p* < 0.01; *** *p* < 0.001.

**Table 6 ijerph-19-04957-t006:** Results of mediation Models 3 and Model 4.

Direct Effect	Indirect Effect	Total Effect
**Model 3**		
Path: Anxiety → Aggressive Behaviors → Adherence to sleep–wake routines		
β = −0.034	β = −0.401 **	β = −0.389 **
**Model 4**		
Path: Anger → Aggressive Behaviors → Adherence to sleep–wake routines		
β = −0.180	β = −0.270 **	β = −0.449 **

Note: * *p* < 0.05; ** *p* < 0.01; *** *p* < 0.001.

## Data Availability

Data are available from the first authors upon reasonable request.

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
