# Peer review of "The Effect of Sleep–Wake Routines on the Negative Emotional States and Aggressive Behaviors in Adults with Autism Spectrum Disorders (ASD) during the COVID-19 Outbreak"

_ijerph, 2022, doi:10.3390/ijerph19094957_

Round 1

Reviewer 1 Report

The manuscript entitled "The effect of the Sleep-wake Routines on the Negative Emotional states and Aggressive Behaviors in Adults with Autism Spectrum Disorders (ASD) during the COVID19 outbreak" by Levante et al. has been well presented here. The authors need to address the minor comments are mentioned below.

  1. The last part of Tables 1 and 3 show the same information. Clarify why the same part was used in these two tables or any other description. 
  2. Since the sample size is small, the authors need to move most of the conclusion text under the limitation title, which will help the readers understand very well about the present study. 
  3. For better understanding, tables 1- 3 can be better presented with three columns, or Gender, Severity, etc., can be moved to the left side of the table. 
  4. Percentage symbols are missing in tables 1 (2(4.7)) and 3 (2(4.7)).

Author Response

Author’s reply: we thank the reviewer for his/her revisions and suggestions. You found all revisions in red in the text of the revised paper.

1. The last part of Tables 1 and 3 show the same information. Clarify why the same part was used in these two tables or any other description. 

Author’s reply: in the revised paper, as footnote, we specified that the info regarding hours spent using technological devices was parent- (table 3) or adult with ASD- reported (table 1).

2. Since the sample size is small, the authors need to move most of the conclusion text under the limitation title, which will help the readers understand very well about the present study. 

Author’s reply: we entitled the last section as Limitation and Conclusion.

3. For better understanding, tables 1- 3 can be better presented with three columns, or Gender, Severity, etc., can be moved to the left side of the table. 

Author’s reply: sorry, we revised the table formatting.

4. Percentage symbols are missing in tables 1 (2(4.7)) and 3 (2(4.7)).

Author’s reply: we add the % symbols. Thank you.

Reviewer 2 Report

Dear Authors,

Thank you for taking up a very interesting topic of „The effect of the Sleep-wake Routines on the Negative Emotional states and Aggressive Behaviors in Adults with Autism Spectrum Disorders (ASD) during the COVID19 outbreak”.  Certainly, the situation of adults with ASD is still underexplored. However, the paper needs significant improvements.

  • The number of keywords is quite large (10 keywords). Usually, the authors limit to several keywrods.
  • More details on sample selection are needed. From the description, I can guess that the research sample was non-radom, but it should be clearly indicated what was the sampling method. Was it the snowball method or maybe the authors reached research participants on Facebook groups?
  • There is a mistake in Table 1. Data are entered twice in the item 'Hours spent using screen-based devices (before lockdown)?’ Probably in the second case, it was about "DURING" the lockdown.
  • Table 2 - "Parental characteristics" shows the percentage of people within the self-identification as a "smart worker". I have doubts whether the respondents identified themselves as smart workers in the same way or not. Have the authors proposed any definition of what a smart worker is?
    This is not a key comment in the context of the subject of the study, but it is worth paying attention to the precision of conceptual definitions.
  • In the case of measures such as "aggressive bahaviors" and "behavioral problems" /lines 144-162/, the choice of scale in the questions is questionable. Why was a scale from 0 to 7 used (with an even number of answer variants and no middle value)? In the case of the two earlier measures "adherence to sleep-wake conditions" and negative emotional states ", the scale was odd (five-point scale).
  • The article suffers from a serious shortcoming in its logical structure. Why do the authors not clearly formulate their hypotheses immediately after the literary review? The first mentions of hypotheses appear only in the "discussion". It is confusing to the reader.
  • There is a lack of the clear presentation of study contributions (both practical and theoretical).
  • The important limitation of the study results was not indicated, which is that due to the non-random sample selection procedure the results are not representative of the study population.
  • In the "discussion" section it is worth to refer to implications and limitations of the study, and the "conclusion" section should actually translate into a short conclusion of the research results.

Best regards,
The reviewer.

Author Response

Dear Authors,

Thank you for taking up a very interesting topic of „The effect of the Sleep-wake Routines on the Negative Emotional states and Aggressive Behaviors in Adults with Autism Spectrum Disorders (ASD) during the COVID19 outbreak”.  Certainly, the situation of adults with ASD is still underexplored. However, the paper needs significant improvements.

Author’s reply: we thank the reviewer for his/her revisions and suggestions. You found all revisions in red in the text of the revised paper.

  • The number of keywords is quite large (10 keywords). Usually, the authors limit to several keywrods.

Author’s reply: we remove three keywords.

  • More details on sample selection are needed. From the description, I can guess that the research sample was non-radom, but it should be clearly indicated what was the sampling method. Was it the snowball method or maybe the authors reached research participants on Facebook groups?

Author’s reply: thank you for the precious suggestions; yes, we recruited participants through snowball method. In other words, we share the online survey link via posts on FB groups or WhatsApp chat and we asked to complete and spread the link. We specified the collection data method in the Method section.

  • There is a mistake in Table 1. Data are entered twice in the item 'Hours spent using screen-based devices (before lockdown)?’ Probably in the second case, it was about "DURING" the lockdown.

Author’s reply: sorry, in the ms we did not included a main info: i.e., in Table 1 the info were adult with ASD reported, whereas in the table 3 these info were parent reported. We added a footnote which specified that.  

  • Table 2 - "Parental characteristics" shows the percentage of people within the self-identification as a "smart worker". I have doubts whether the respondents identified themselves as smart workers in the same way or not. Have the authors proposed any definition of what a smart worker is?
    This is not a key comment in the context of the subject of the study, but it is worth paying attention to the precision of conceptual definitions.

Author’s reply: we better explain you our question included in the survey: we ask to parents if they have had continued to work at home during the lockdown. In the revised paper, we modified the name of the variable in order to be more understandable by the readers. We hope this would be more clarify.

  • In the case of measures such as "aggressive bahaviors" and "behavioral problems" /lines 144-162/, the choice of scale in the questions is questionable. Why was a scale from 0 to 7 used (with an even number of answer variants and no middle value)? In the case of the two earlier measures "adherence to sleep-wake conditions" and negative emotional states ", the scale was odd (five-point scale).

Author’s reply: We try to better explain you the response options used in our survey.

Regarding the measures “adherence to sleep-wake” and “emotional states” we apply a 5-point Likert scale in order to evaluate the frequency of the parent perception of the child adherence to the routines and emotional states. To be specific, the response options varied from “not at all” to “less” to “enough” to “much” to “very much”. We think that this scale adequately served to our purposes and it were used in previous studies (see Bianco et al., 2021; Levante et al., 2021; Petrocchi et al., 2020).

Regarding the “aggressive behaviors" and "behavioral problems” measures, our purposes were to:

1) provide a broadly response options range to parents;

2) obtain a polarized parent-reported answer regarding the behavioral problems.

The last two scales were conceived taking a cue from other standardized scales, e.g., CBCL in which the middle value is not considered. Furthermore, aggressive behavior measure was already applied in a published paper (Levante et al., 2021).

We are awareness that our measures were not standardized; indeed we reported this issue as limitation in the last section of the paper. You found the sentence in red in the text.  

  • The article suffers from a serious shortcoming in its logical structure. Why do the authors not clearly formulate their hypotheses immediately after the literary review? The first mentions of hypotheses appear only in the "discussion". It is confusing to the reader.

Author’s reply: we thank you for the suggestions. We better clarify the HPs formulated in the revised manuscript. We hope this revised section was more readable.  

  • There is a lack of the clear presentation of study contributions (both practical and theoretical).

Author’s reply: regarding this suggestions, we added two sentence in introduction section in which we specified the study contributions. A further sentence were added in the abstract. Finally, in the discussion section, we highlighted in red the sentences already reported in the paper in which we reported the study contributions.

  • The important limitation of the study results was not indicated, which is that due to the non-random sample selection procedure the results are not representative of the study population.
  • Author’s reply: we add this limitation in the Limitation and Conclusion section.
  • In the "discussion" section it is worth to refer to implications and limitations of the study, and the "conclusion" section should actually translate into a short conclusion of the research results.

Author’s reply: according to the previous reviewer, we entitle the last section as Limitation and Conclusion. Nevertheless, we are available to further revise the structure of the section in according to all reviewers suggestions.

Best regards,
The reviewer.

Reviewer 3 Report

The study is well written, the illustrated methodology is clear, the type of analysis appropriate, and the results and discussion are consistent with the purpose of the work. Here are some suggestions:

  • line 19-20 is not clear, please reformulate
  • in the abstract no numeric results are reported, please complete the abstract with part of the numeric findings
  • line 37-38 need some English revision
  • please specify date and university of ethical approval and the protocol adherence to  the Declaration of Helsinki
  • table 3 and 1 repeated some of the reported results, please correct
  • table 1-2-3 and participants characteristics are results and should be reported in results section
  • lines 135-137, 148-149 intend to report the creation of a dummy variable and should be moved in the statistical analysis section

Author Response

The study is well written, the illustrated methodology is clear, the type of analysis appropriate, and the results and discussion are consistent with the purpose of the work. Here are some suggestions:

Author’s reply: we thank the reviewer for his/her revisions and suggestions. You found all revisions in red in the text of the revised paper.

  • line 19-20 is not clear, please reformulate

Author’s reply: done, thank you.

  • in the abstract no numeric results are reported, please complete the abstract with part of the numeric findings

Author’s reply: done, thank you.

  • line 37-38 need some English revision

Author’s reply: done, thank you.

  • please specify date and university of ethical approval and the protocol adherence to  the Declaration of Helsinki

Author’s reply: this info was reported after the Conclusion section, in the Institutional Review Board Statement; however, we add the sentence in the procedure section.

  • table 3 and 1 repeated some of the reported results, please correct

Author’s reply: we added a footnote which specified that in table 1 these info were adult with ASD reported, whereas in the table 3 these info were parent reported.

  • table 1-2-3 and participants characteristics are results and should be reported in results section

Author’s reply: as you stated, data included in tables could be considered as results. Nevertheless, we think that they would be useful to describe the sample recruited, their characteristics, and routines. Therefore, we inserted these information regarding the sample characteristics in the participants section. Furthermore, we did not included them in mediation analysis. We hope you understand our conceptualization of the paper structure.  

  • lines 135-137, 148-149 intend to report the creation of a dummy variable and should be moved in the statistical analysis section

Author’s reply: we are sorry, but by the lines indicated by you we did not understand your request. The lines 135-137 and 148-149 corresponded to the description of measures. Could you give us specific information in order to answer to your revision? Thank you.

Round 2

Reviewer 2 Report

Dear Authors,

Thanks for referring to my remarks. In my opinion, the quality of the article in its current form has improved, so I can recommend it to the publication.

Best regards,
The Reviever